# Lumbar Paravertebral Muscle Pain Management Using Kinesitherapy and Electrotherapeutic Modalities

**DOI:** 10.3390/healthcare12080853

**Published:** 2024-04-18

**Authors:** Sînziana Călina Silişteanu, Elisabeta Antonescu, Lavinia Duică, Maria Totan, Andrei Ionuţ Cucu, Andrei Ioan Costea

**Affiliations:** 1Faculty of Medicine and Biological Sciences, Stefan cel Mare University of Suceava, 720229 Suceava, Romania; sinziana.silisteanu@usm.ro (S.C.S.); andrei.cucu@usm.ro (A.I.C.); andrei.costea@usm.ro (A.I.C.); 2Faculty of Medicine, Lucian Blaga University of Sibiu, 550169 Sibiu, Romania; maria.totan@ulbsibiu.ro

**Keywords:** low back pain, therapeutic physical exercises, mobility, pain

## Abstract

Background: Low back pain is considered a public health problem internationally. Low back pain is a cause of disability that occurs in adolescents and causes negative effects in adults as well. The work environment and physical and psychosocial factors can influence the occurrence and evolution of low back pain. Methods: The purpose of this paper is to highlight the physiological and functional changes in young adults with painful conditions of the lumbar spine, after using exercise therapy. The study was of the longitudinal type and was carried out over a period 6 months in an outpatient setting. The rehabilitation treatment included electrotherapeutic modalities and kinesitherapy. Results: The results obtained when evaluating each parameter, for all moments, show statistically significant values in both groups. The results obtained regarding the relationship between the therapeutic modalities specific to rehabilitation medicine and low back pain are consistent with those reported in studies. Conclusions: Depending on the clinical-functional status of each patient, kinesitherapy can accelerate the heart rate and increase the blood pressure and oxygen saturation of the arterial blood, values that can later return to their initial levels, especially through training.

## 1. Introduction

Low back pain (LBP) is considered a public health problem internationally. LBP affects 70–85% of the population and accounts for 2.3% of physician visits, being the most common part of the body that presents with pain [1]. LBP is the most common musculoskeletal disorder [2,3] affecting people of all ages [4], but for young adults (20–29 years), it is easy to identify the role of socio-demographic factors and lifestyle in the occurrence of LBP [5,6]. LBP is a cause of disability that occurs in adolescents and causes negative effects in adults as well [7].

Many causes of LBP are degenerative changes in the intervertebral discs. This aspect is a consequence of changes in nutrition processes at the level of the intervertebral discs that cause circulation problems, reducing the permeability of the vertebral body, reducing the transport of nutrients, accumulating catabolic final products, and decreasing tissue pH by activating enzymes that play a role in disc matrix degradation. Lumbosacral spine dynamics require stability and mobility. Stability is provided by the vertebral bodies and intervertebral discs, while mobility is provided by the facet joints. Mechanical, postural and genetic factors are involved in the onset of LBP [8,9].

From an anatomical point of view [10,11,12], chronic and severe low back pain can originate from intervertebral discs (degeneration and disc herniation), central and foraminal stenosis and facet joints, but also from the sacroiliac joints. In the context of mechanical dysfunction associated with LBP, relevant studies [13] emphasize the importance of translation (anterior/posterior) and intervertebral laxity. The kineto-pathological model highlights the fact that LBP results from the repeated use of stereotyped movements of a movement direction (flexion–extension, rotation, lateral inclinations or combinations). The typical model is the one in which, during the execution of a movement (flexion) or assuming a prolonged position (sitting), the lumbar spine mobilizes in its available range in one direction. Repetition of the same movement contributes to the strain on the lumbar spinal tissue. Over time, the stress accumulation rate is higher than the adaptive tissue remodeling. Thus, tissue irritation, lumbar pain, and microlesions occur, leading to persistent LBP [11].

There are studies that highlight, from a pathological point of view, the role of mediators released from the herniated nucleus pulposus in the production of compressive lesions of disc herniation or stenosis [13], the increase in the number of CGRP-type neurons (molecular marker for the peptidergic population of primary afferent nociceptors) [14] that are responsible for discogenic pain. Pain-related functional behavior is, on the one hand, the motor response to pain or pain-related suffering, and on the other hand, the effect of root compression on the severity of paraspinal muscle degeneration, and is characterized by low mobility of the lumbar spine [14,15], the impairment of muscle contraction [15] and the inability to achieve trunk flexion [16,17]. Moissenet’s study [18] showed that for 85–90% of patients, the causes of low back pain cannot be established with certainty, the patients being classified as having non-specific back pain. Among them, approximately 10% have chronic back pain, thus becoming a socio-economic burden [19].

LBP is considered a complex condition in which central and peripheral nociceptive processes are influenced by various factors (social, psychological, musculoskeletal) that interact with each other [20,21]. Like many other conditions, back pain is influenced by genetic and environmental factors [22,23,24,25]. That is why recovery programs involve the evaluation of biomarkers that provide objective indications of the patient’s condition, and that can be measured accurately, the data being reproducible [20]. The meta-analysis by Shiri [26] assessed the association between overweight/obesity and LBP. 

The work environment and physical and psychosocial factors can influence the occurrence and evolution of LBP. Ergonomic risk factors lead to LBP, while psychosocial factors may influence the disability component of LBP. Prolonged posture causes static loading of soft tissues and causes dysfunction, while excessive standing and sitting can cause circulation problems, energy consumption, reduced mobility, accumulation of metabolites, accelerated disc degeneration and then disc herniation [27].

However, different studies indicate that pain can be impacted by lifestyle factors such as sleep disruptions [28,29], physical activity levels [30], sedentary behavior, and physical deconditioning [31]. Additionally, sleep disorders are linked to musculoskeletal health [29,32]. Consequently, engaging in physical activity not only helps maintain musculoskeletal health but also benefits cardiovascular health, reduces cancer risk, and lowers overall mortality rates [33,34]. Therefore, back pain, which is widely reported as the most common musculoskeletal issue globally [35], is strongly linked to sleep quality [36], disability [37], anxiety, and depression. Poor sleep quality exacerbates musculoskeletal pain intensity [38,39,40,41,42,43]. Research [44,45] also indicates that traditional exercise programs notably enhance sleep quality in patients with musculoskeletal disorders.

Other studies demonstrate that patients with spinal disorders presented several factors of poor prognosis, such as the presence of chronic conditions [46,47], severe initial pain [48], the influence of mental status and, last but not least, female sex [49], in which it seems that sleep disorders are more evident [50,51,52], sleep being one of the adjustable risk factors [46]. Currently, exercise-based physical therapy is the primary initial treatment for musculoskeletal disorders [44,53,54,55] and improves sleep quality. A meta-analysis by Kelley et al. [56] demonstrated that exercise significantly improved overall sleep quality, as did a recent randomized controlled trial by Tseng et al., who demonstrated that moderate-intensity exercise has a significant beneficial effect on sleep quality and cardio-autonomic function [57,58].

Guidelines recommend non-pharmacological and non-invasive management [59], which include patient education and the application of exercise therapy. The guidelines regularly recommend the practice of therapeutic physical exercises for non-specific LBP [60,61], psychological or cognitive behavioral therapies, yoga, with the benefits being observed by the assessment of pain and functional dysfunction over 3 months or even earlier [62]. Physical exercises, alone or accompanied by patient education and advice, have been shown to be effective for preventing LBP [63]. 

Lumbar stabilization exercise programs (LSEP) aim to control and coordinate the paraspinal and abdominal musculature [64]. Performing physical exercises aims to improve the postural muscles, as they have a role in stabilization and neuro-coordination or their combination [65]. Macedo’s study [66] showed the effectiveness of motor control exercise compared to other exercises for the lumbosacral segment of the spinal cord [31].

Exercise is an effective way to treat back pain. Physical exercise lasting 30 min can be performed by every patient and is useful to decrease pain, increase the flexibility of the spine and also increase the quality of life. Physiotherapy influences the circulatory system and the myo-arthro-kinetic system, which indicates, at the level of the central nervous system through receptors, every time, the position of the different segments of the body, while at the cortical level, the order of execution and control is elaborated for each motion. Physical therapy reduces pain and improves balance, coordination and joint flexibility. Each exercise program intensifies local catabolism, while the resulting substances are biologically stimulated to intensify anabolism, and on the other hand, muscle tone is influenced.

Physical activity, by involving the muscles, will improve the metabolism and allow a large increase in the amount of O_2_ in the body, taking into account the fact that the muscle is the main producer of lactic acid and CO_2_ and the source of heat to the body. Correct learning of some movements will contribute to better coordination of readily automated movements in the form of unconditioned motor reflexes that, together with conditioned reflexes, will contribute to the formation of automatism, which means that movements can be performed without the need for immediate control by the cortex. Programming these motor skills can create points of excitation and inhibition in the central nervous system that will cause muscles to contract in a certain sequence and relax in others [67,68]. 

Therapeutic ultrasound is a form of physiotherapeutic treatment that uses high-frequency sound waves, which have ability to penetrate tissues at different depths depending on the frequency of application. Ultrasound can be applied in continuous form, having especially a thermal effect that causes muscle relaxation, and in pulsatile form, in which the thermal effect is diminished, improving blood circulation and reducing pain. The biological effects of ultrasound are mild, reversible and irreversible, even destructive, depending on the intensity, frequency, amplitude and duration of application [69,70]. Therapeutic ultrasound can contribute to the management of lumbar pain by reducing inflammation, due to the stimulation of blood circulation, reducing the sensation of pain associated with inflammation by relaxing tense muscles, and reducing discomfort by healing tissues, due to the increase in circulation and the improvement of local metabolism. There are few studies to support the use of ultrasound in patients with LBP to reduce pain and improve the quality of life of these patients. The improvement of the symptomatology through the application of ultrasound is of short duration. Studies are needed to estimate the effect of ultrasound in reducing pain, improving physical function and increasing the quality of life of patients with LBP [71]. 

Transcutaneous electrical nerve stimulation (TENS) is a non-pharmacological type and is used to treat acute and chronic pain. TENS activates inhibitory mechanisms to reduce CNS excitability and reduce pain. TENS trains large peripheral afferent fibers, and inhibitory systems are activated that reduce analgesia. The analgesic effect of TENS is mediated by receptors sensitive to a certain frequency. Thus, at a low frequency of 2–5 Hz, endorphins are stimulated, providing a calming and pain-reducing effect. At the high frequency of 80–100 Hz that is frequently used, TENS causes analgesia by activating some inhibitory mechanisms at the CNS level, involving opioid receptors, GABA, and serotonin. Repeated daily applications of TENS using the same parameters (intensity, duration, frequency) can produce analgesic tolerance in a few days [72]. There are clinical studies that recommend TENS as an adjunctive treatment for osteoarthritis or rheumatoid arthritis [73], in adults with hip fracture when no other electrotherapy applications are performed for the pain [74], and for low back pain moderate in intensity, together with physical exercises and NSAID administration [6,75].

The laser is a technological device that emits coherent electromagnetic radiation through the process of stimulating the emission of radiation. The operating principle of a laser is based on the properties of laser radiation: coherence, directionality and amplification. Low-level laser therapy (LLLT) is a form of treatment that involves the use of a lower-power laser to influence biological processes in the treated tissues. In the treatment of low back pain, low-intensity laser therapy can have several beneficial roles: analgesia, reduction of inflammation, stimulation of blood circulation, stimulation of healing processes, and muscle relaxation. LLLT is an alternative therapy to pharmacological treatments for chronic pain. Despite its widespread use, the effectiveness of LLLT is still controversial [76].

This study aimed to emphasize physiological and functional alterations in the paravertebral muscles of young adults experiencing lumbar spine pain following the use of electrotherapy combined with therapeutic kinetic exercises. 

The initial hypothesis questioned whether significant differences due to engaging in therapeutic kinetic exercises and receiving electrical currents existed in young adults diagnosed with low back pain.

## 2. Materials and Methods

### 2.1. Study Description

This was a prospective longitudinal study carried out over a period of 6 months in an outpatient setting. In general, before the start of recovery treatment, the patient is informed about its content and signs a consent form. The study was approved by the Research Ethics Commission within the USV with No. 31/19 April 2021.

Two hours before the start of the treatment, participants in this study were advised to refrain from physical exercise, smoking and intake of caffeinated beverages.

For this study, only some of the patients who came for medical control agreed to participate. Therefore, the study finally included a number of 175 patients (aged between 20–40 years) who presented to the medical office with pain in the lumbosacral spine (Table 1). The patients were evaluated clinically, paraclinically, functionally and by imaging techniques at the beginning of the treatment (T1—before the start of the treatment), at the end of it (T2—after 15 days of application of the recovery treatment) and at the control (T3—after 20 weeks after the end of treatment).

Pain was assessed according to the visual analog scale (VAS), in which the patient selected a whole number that best showed the intensity of his pain. The VAS scale had values between 0 (no pain) and 10 (maximum unbearable pain). The Oswestry Index was applied to assess the disability of patients with low back pain, assessing the impact the pain has on the patients’ daily life sectors. It contains 10 questions and has values between 50 (severe pain and impairment of functional capacities) and 0 (minimal pain and minimal disability). The Oswestry Disability Index (ODI) is a well-known and widely used tool in the assessment of disability related to low back pain and is the most common patient-reported outcome for assessing disability due to back pain.

The assessment of functionality and quality of life of patients with painful conditions of the lumbar spine was performed using the LBP-Module scale. It contains 10 questions (0 = minimum level of functionality and quality of life and 30 = maximum level of functionality and quality of life).

The finger-to-floor distance index (FFDI) (normal = 0 cm) was used to assess the mobility of the lumbar spine. With the help of measuring tape, the distance between the top of the medius and the ground was estimated after the patient flexed the trunk on the thighs, starting from the orthostatic position. 

The physiological parameters monitored in the patients were pulse, blood pressure (both parameters show the tested person’s ability to adapt to the effort), and peripheral blood oxygen saturation (by oximetry). The blood oxygen level is actually the percentage of oxygen carried by red blood cells, and normal oxygen level values are considered to be between 95–100% [77,78].

The evaluation was performed in the physical therapy room at a suitable temperature that did not influence the sensor and, implicitly, the measurement. For the accuracy of the results, the pulse oximeter was applied to the index finger, while for women, nail polish or fake nail applied on the finger was removed. This parameter required the use of the pulse oximeter, a validated medical device that measures peripheral oxygen saturation (Sp O_2_) and pulse. Blood oxygen saturation was monitored, and the measurement accuracy was ±2% for pulse and ±2% for oxygen. Blood pressure with its two components (systolic and diastolic) was assessed using a Rossmax digital blood pressure monitor, a validated medical device [79,80,81]. 

The evaluation was carried out with the left arm in support and the patient in a resting position, seated on a chair and supported by a backrest [82,83]. 

Recovery treatment included electrotherapy and kinesitherapy (therapeutic physical exercises).

The applied electrotherapy had an analgesic, sedative, anti-inflammatory role, with frequencies between 50–500 Hz and intensity below the motor threshold for a muscle relaxation effect. TENS, ultrasound and laser have been used. Thus, for the acute forms, the LLLT 3 J/cm^2^ laser was used with a power density between 0.05–5 W/cm^2^ (anti-inflammatory analgesic effect, combating muscle contraction, decreasing joint stiffness, increasing joint amplitude). For the analgesic and stimulating effect, TENS was also used, created on the principle of “pain deviation” or “gate control theory”. The applied parameters were the intensity of 100 mA, the duration of 150 μs, and the frequency of 50–100 imp/s (Hz). Ultrasound has been used for analgesic, anti-inflammatory, fibrinolytic effects and for changing cell membrane permeability. The pulsating form was used to reduce the deep thermal effect, as happens in the continuous form. The effects were for acute and chronic conditions, causing an increase in the pain threshold, an increase in local blood circulation, and a decrease in joint stiffness. The applied frequency was 1 MHz, the intensity was 0.3 W/cm^2^, and the duration was 4 min. The vibrations determined by the mechanical oscillations have a high-frequency micro massage action, while the ultrasonic oscillations stimulate diffusion processes at the membrane level by intensifying metabolism and tissue regeneration processes. Ultrasound penetration was 3–6 cm.

The objectives of the applied physiotherapy were as follows:Improving mobility of the lumbosacral spine affected by paravertebral muscle contracture and instabilityCorrection of static vertebral disordersRestoring the strength of the paravertebral muscles and gluteal and abdominal muscles by ensuring stability and dynamics of the lumbosacral spine.Correction of motor deficit in radiculopathy (coordination, strength, resistance)Re-education of orthostatism and gait

Postural control, passive and active mobilization exercises, and static (isometric) and dynamic resistance exercises were used. Useful Williams techniques were applied to reduce pain: to increase stability of the lower train, to ensure flexion–extension balance for postural muscles, to tone abdominal muscles, and to ensure pelvic tilt. Basically, the Williams program was used to flex the lower trunk and included exercises that tried to remobilize the lumbar spine, allowing pelvic tilts, stretching the psoasiliac muscle, but also the paravertebral muscles. The exercises were performed during the daily treatment period for 45 min/session in the physical therapy room.

The program included the following:Warm-up period of 8–10 min, lumbar spine stretching exercises, breathing exercises, and mobilization of the body segments towards the trunk were applied.The actual program that started with 15 min and increased to 25 min, with aerobic exercises of medium intensity, bicycle, stepper, and walking on the treadmill. Maintaining a correct posture was followed, and exercises were performed to maintain proprioceptive capacity using specific devices.The final period with a duration between 5–10 min, in which light exercises were performed to avoid brutal exertional hypertension, as well as relaxed walking exercises and breathing exercises.

Upon completion of the treatment, participants maintained a home program of therapeutic exercises, conducting three sessions weekly, each lasting at least 30 min. They were provided with verbal instructions and written materials and encouraged to continue the exercises independently in the kinesitherapy room: The Back School [84,85]. It is important to emphasize that the effectiveness of their home exercise program hinges on maintaining proper ergonomics during these sessions. Ensuring ergonomic practices is crucial for preventing potential discomfort or injuries during home exercises.

Participants were educated on how their home environment and daily activities could impact their posture and musculoskeletal health. They were provided with specific guidance on optimizing ergonomics, such as maintaining proper sitting positions, using supportive furniture, and creating a conducive workspace. This emphasis on ergonomics aims to enhance the effectiveness of the therapeutic exercises and ensure a seamless transition from the rehabilitation facility to the home setting.

During their follow-up in the rehabilitation program, we evaluated the effects of both the therapeutic exercise and the incorporation of ergonomic principles into their daily activities. This holistic approach not only attends to immediate rehabilitation needs but also fosters long-term musculoskeletal health and well-being for the participants.

The Williams program comprises flexion training and aims to unload the lumbar area and reduce disc-radicular conflict. The Williams method can be practically expressed in three phases: In phase 1 (acute phase), 6 exercises are carried out with the aim of increasing the mobility of the lower trunk, toning the abdominal muscles, and lengthening the muscles of the lumbosacral spine and the one in the posterior region of the thigh; 5 of these exercises are performed from the supine position, and one from the sitting position. In phase 2 (subacute phase), the exercises are performed after 2 weeks and include exercises that are executed from free positions, from hanging on a fixed ladder and moving the lower limbs. In phase 3 (chronic stage), exercises include tilting the pelvis, stretching the hip flexors, and toning the muscles of the trunk, abdomen, and buttock to maintain a neutral position of the pelvis. Each exercise was performed 3–5 times depending on the patient’s ability and own clinical-functional conditions.

Examples from the flexion program include pelvic balance, knees to chest, then to chest (all supine), partial raise, hamstring stretch, and hip flexors.

### 2.2. Presentation of the Group

All participants (patients and evaluators) in the study were informed of the purpose of the study and the exercise program, the collected data being used only for this study.

The original group of 175 patients was randomly divided into two groups: the first 86 participants, in order of registration, were assigned to group G1, and the next 89 participants to group G2. Each group received electrotherapy, and one of the groups also received kinetic therapy. Thus, group G1 (which included kinetic therapy in the treatment) included 86 patients, of whom 44 (51.16%) were women and 42 (48.84%) were men. Group G2 included 89 patients, of whom 45 (50.56%) were women and 44 (49.44%) were men (Table 2).

The participants in the study who required days of medical leave benefited from a program of therapeutic exercises aimed at improving their health and facilitating their return to work. At home, the therapeutic exercise program becomes crucial because it contributes to the rehabilitation of participants who have left their daily routine.

The study participants who continued their work at the workplace managed their work responsibilities alongside the rehabilitation program. The program of therapeutic exercises at home was integrated into the daily routine and designed to adapt to their work schedules.

Ergonomic factors are particularly crucial for participants who are working while undergoing rehabilitation. Implementing proper ergonomics in their workplaces can mitigate additional strain and foster a healthier work environment.

It is essential to consider these factors when designing and evaluating rehabilitation programs, as they may influence participant adherence to the home exercise regimen and the overall success of the intervention. In addition, understanding the participants’ working conditions provides a valuable context for interpreting the experimental results in the context of real-world applications and everyday life scenarios.

After completing the initial treatment, participants were categorized into distinct clusters based on their occupations: students, teachers, drivers, individuals from the marketing field and daily laborers. Each cluster continued the therapeutic exercise program at home. 

Both verbal instructions and written materials were provided, specifically tailored to address the unique needs and challenges associated with each occupation.
Student Cluster:
-Received a set of therapeutic exercises focusing on ergonomics in the workplace.-The home exercise program for students focused on addressing the impact of prolonged sitting during study sessions.-Exercises emphasized posture improvement and stretches to alleviate tension from prolonged desk work or computer use.Teacher Cluster:
-Received a set of therapeutic exercises focusing on ergonomics in the workplace.-Teachers, who often face demands related to standing for extended periods and repetitive motions, received a tailored program.-Their exercises aimed to enhance strength and flexibility, particularly in areas affected by frequent standing and classroom activities.Driver Cluster:
-Participants in the driver cluster engaged in exercises designed to counteract the effects of prolonged sitting and repetitive movements associated with driving.-The program included stretches and strengthening exercises targeting areas vulnerable to strain during driving.Marketing Field Cluster:
-Participants from the marketing field engaged in a home exercise program designed to address the sedentary nature of office work common in marketing roles.-Exercises focused on mitigating the effects of prolonged sitting, promoting good posture and targeting areas prone to tension from desk work.Daily Laborer Cluster:
-Daily laborers, who often engage in physically demanding tasks, received a tailored program aimed at improving strength, flexibility, and resilience.-Exercises were designed to address the physical demands and potential strain associated with manual labor, promoting overall musculoskeletal health.Other occupations with varied health conditions:
-Followed a comprehensive home exercise program, combining elements of both ergonomics and general musculoskeletal rehabilitation.-Received personalized instructions to address specific health conditions or concerns within this diverse cluster.

Regular check-ins and feedback sessions were conducted with each cluster to assess adherence and address any concerns or modifications needed based on individual responses. The participants returned for a comprehensive evaluation in the rehabilitation program, where outcomes within each cluster were analyzed. This assessment included improvements in pain levels, functional capacity, and overall well-being. 

Statistical analyses were performed to determine the significance of the interventions within each occupational cluster, providing insights into how individuals in the marketing field and daily laborers responded to their tailored therapeutic exercises.

This approach allowed for a nuanced understanding of the effectiveness of the program in addressing the specific occupational challenges faced by individuals in the marketing field and daily laborers, contributing to a more comprehensive evaluation of the rehabilitation program. 

Throughout the intervention period, participants were encouraged to maintain adherence to their prescribed home exercise program. 

Upon returning for further evaluation in the rehabilitation program, the outcomes for patients within each cluster were meticulously assessed. This included analyzing improvements in pain levels, functional capacity, and overall well-being. 

The average age of patients in the G1 group was 31.05 years (±5.48), and in the G2 group, it was 29.55 years (±5.51) By age group, it was found that the most affected were those in the 31–35 year-old group, with women being more affected in both groups. 

### 2.3. Statistical Analysis

For statistical analysis, categorical variables were expressed as number and percentage, while continuous variables were expressed as mean ± standard deviation. The significant differences between the two groups in terms of the scales used at the time of evaluation were evaluated with the help of the independent samples t test. The interpretation of the *p* values was as follows: *p* < 0.05, the statistical link was significant (95% inter-wave confidence) and if *p* > 0.05, the statistical relationship was insignificant. Statistical analyzes were performed with SPSS 20 (SPSS Inc., Chicago, IL, USA).

## 3. Results

In our study, we evaluated the effectiveness in relieving pain and improving the quality of life in two groups of patients (G1 and G2) at different time points (T1, T2, T3): 

PAIN parameter (VAS) 

In the G1 group, the pain assessed using the VAS scale decreased by 31.79% at T2 and by 24.57% at T3.In the G2 group, the improvement was 27.49% at T2 and 25.68% at T3.Between T1 and T3, pain relief was 56.35% in the G1 group and 53.17% in the G2 group.

Quality of life parameter (LBP-M) 

In the G1 group, the functional evaluation and the patient’s quality of life evaluated using the LBP-Module scale showed an increase of 29.58% at T2 and 35.93% at T3.In group G2, the increase was 30.72% at T2 and 32.80% at T3.Between T1 and T3, the increase in quality of life was 65.52% in the G1 group and 63.54% in the G2 group (Appendix A).

DISABILITY parameter—Oswestry Index

In the G1 group, the disability assessed using the Oswestry scale was reduced by 29.15% at T2 and by 21.43% at T3.In the G2 group, the reduction was 19.02% at T2 and 17.55% at T3.The disability assessed between T1 and T3 decreased by 50.55% in the G1 group and by 36.57% in the G2 group.

FINGER-TO-FLOOR DISTANCE INDEX (FFDI) parameter 

In the G1 group, the finger-to-ground index (FGI) decreased by 37.56% at T2 and by 32.5% at T3.In the G2 group, the decrease was 10.62% at T2 and 7.04% at T3.Between T1 and T3, a reduction of 69.98% in the G1 group, compared to 17.6% in the G2 group, was found (Appendix A).

HEART RATE (HR) parameter 

In the G1 group, heart rate improved by 12.44% at T2 and by 1.63% at T3.In the G2 group, the improvement was 2.32% at T2 and 7.26% at T3.Between T1 and T3, the heart rate improved by 10.81% in the G1 group compared to 9.58% in the G2 group.

OXYGEN SATURATION (SAT O_2_) parameter

In the G1 group, the obtained results indicated an increase of 0.75% at T2 and 0.2% at T3.In the G2 group, the increase was 0.38% at T2 and 0.17 at T3.Between T1 and T3, there was an increase in blood oxygen saturation of 1.28% in group G1 and 0.2% in group G2 (Appendix A).

SYSTOLIC BLOOD PRESSURE (SBP) parameter 

In the G1 group, SBP registered a change, namely a reduction of 11.11% at T2 and of 0.61% at T3.In the G2 group, the reduction was 10.74% at T2 and 2.09% at T3.Between T1 and T3, TAS decreased by 10.49% in the G1 group and by 8.39% in the G2 group.

DIASTOLIC BLOOD PRESSURE (DBP) parameter 

In the G1 group, TAD registered a decrease of 13.56% at T2 and 10.49% at T3.In group G2, the decrease was 4.65% at T2 and 4.32% at T3.Between T1 and T3, TAS decreased in the G1 group by 8.9% and in the G2 group by 6.17% (Appendix A).

Regarding the subgroups of women from study groups G1 and G2, we can state the following: 

PAIN parameter (VAS) 

In the G1 group, the pain assessed using the VAS scale decreased by 32.64% at T2 and by 25.06% at T3.In the G2 group, the improvement was 26.75% at T2 and 25.45% at T3.Between T1 and T3, pain relief was 57.7% in the G1 group and 52.2% in the G2 group.

Quality of life parameter (LBP-M) 

In the G1 group, the functional evaluation and the patient’s quality of life evaluated using the LBP-Module scale showed an increase of 30.36% at T2 and of 35.31% at T3.In group G2, the increase was 31.07% at T2 and 32.33% at T3.Between T1 and T3, the increase in quality of life was 65.67% in the G1 group and 63.40% in the G2 group (Appendix A).

DISABILITY parameter—Oswestry Index 

In the G1 group, the disability assessed using the Oswestry scale was reduced by 27.84% at T2 and by 21.01% at T3.In the G2 group, the reduction was 18.73% at T2 and 17.37% at T3.The disability assessed between T1 and T3 decreased by 48.86% in the G1 group and by 36.11% in the G2 group.

FINGER-TO-FLOOR DISTANCE INDEX (FFDI) parameter 

In the G1 group, the finger-to-ground index (FGI) decreased by 38.55% at T2 and by 30.18% at T3.In the G2 group, the decrease was 10.78% at T2 and 7.71% at T3.Between T1 and T3, a reduction of 68.74% in the G1 group, compared to 18.40% in the G2 group, was found (Appendix A).

HEART RATE (HR) parameter 

In the G1 group, heart rate improved by 9.59% at T2 and by 1.31% at T3.In the G2 group, the improvement was 2.53% at T2 and 7.12% at T3.Between T1 and T3, the heart rate improved by 8.27% in the G1 group compared to 9.65% in the G2 group.

OXYGEN SATURATION (SAT O_2_) parameter

In the G1 group, the obtained results indicated an increase of 1.16% at T2 and 0.12% at T3.In the G2 group, the increase was 0.37% at T2 and 0.18% at T3.Between T1 and T3, there was an increase in blood oxygen saturation, of 1.29% in group G1 and 0.18% in group G2 (Appendix A).

SYSTOLIC BLOOD PRESSURE (SBP) parameter 

In the G1 group, SBP registered a change, namely a reduction of 11.35% at T2 and of 0.29% at T3.In the G2 group, the reduction was 11.06% at T2 and 2.67% at T3.Between T1 and T3, TAS decreased by 11.06% in the G1 group and by 8.38% in the G2 group.

DIASTOLIC BLOOD PRESSURE (DBP) parameter 

In the G1 group, TAD registered a decrease of 14.80% at T2 and 10.53% at T3.In group G2, the decrease was 5.58% at T2 and 4.56% at T3.Between T1 and T3, TAS decreased by 9.21% in the G1 group and by 5.97% in the G2 group (Appendix A).

Regarding the subgroups of male patients from study groups G1 and G2, the following statements can be made: 

PAIN parameter (VAS) 

In the G1 group, the pain assessed using the VAS scale decreased by 30.64% at T2 and by 24.05% at T3.In the G2 group, the improvement was 27.90% at T2 and 25.45% at T3.Between T1 and T3, pain relief was 56.70% in the G1 group and 27.90% in the G2 group.

Quality of life parameter (LBP-M) 

In the G1 group, the functional evaluation and the patient’s quality of life evaluated using the LBP-Module scale showed an increase of 28.79% at T2 and of 36.60% at T3.In group G2, the increase was 30.23% at T2 and 33.38% at T3.Between T1 and T3, the increase in quality of life was 65.39% in the G1 group and 63.62% in the G2 group (Appendix A).

DISABILITY parameter—Oswestry Index 

In the G1 group, the disability assessed using the Oswestry scale was reduced by 29.15% at T2 and by 21.72% at T3.In the G2 group, the reduction was 19.02% at T2 and 17.55% at T3.The disability assessed between T1 and T3 decreased by 50.87% in the G1 group and by 36.57% in the G2 group.

FINGER-TO-FLOOR DISTANCE INDEX (FFDI) parameter 

In the G1 group, the finger-to-ground index (FGI) decreased by 36.57% at T2 and by 33.19% at T3.In the G2 group, the decrease was 10.16% at T2 and 6.39% at T3.Between T1 and T3, a reduction in the G1 group of 69.76%, compared to 16.94% in the G2 group, was found (Appendix A).

HEART RATE (HR) parameter 

In the G1 group, heart rate improved by 15.19% and by 1.98% at T3.In the G2 group, the improvement was 2.01% at T2 and 9.6% at T3.Between T1 and T3, the heart rate improved by 13.21% in the G1 group compared to 5.78% in the G2 group.

OXYGEN SATURATION (SAT O_2_) parameter 

In the G1 group, the obtained results indicated an increase of 1.16% at T2 and 0.12% at T3.In the G2 group, the increase was 0.37% at T2 and 0.18% at T3.Between T1 and T3, there was an increase in blood oxygen saturation of 1.29% in group G1 and 0.18%in group G2 (Appendix A).

SYSTOLIC BLOOD PRESSURE (SBP) parameter

In the G1 group, SBP registered a change, namely a reduction of 11.35% at T2 and of 0.29% at T3.In the G2 group, the reduction was 11.06% at T2 and 2.67% at T3.Between T1 and T3, TAS decreased by 11.06% in the G1 group and by 8.38% in the G2 group.

DIASTOLIC BLOOD PRESSURE (DBP) parameter 

In the G1 group, TAD registered a decrease of 14.80% at T2 and of 10.53% at T3.In group G2, the decrease was 5.58% at T2 and 4.56% at T3.Between T1 and T3, TAS decreased by 9.21% in the G1 group and by 5.97% in the G2 group (Appendix A).

The results obtained when evaluating each parameter, for all time points, showed statistically significant values in both groups G1 and G2 (Table 3).

And, for the subgroups of women from the two groups G1 and G2, the results were statistically significant (Table 4).

The results were also statistically significant in the subgroups of male patients from groups G1 and G2 (Table 5).

## 4. Discussion

The results showed changes between the evaluation time points, such that in the evaluations between T2 and T3, the patients in the G2 group registered a decrease in the values obtained for the evaluated parameters, which also influenced the values at T3. The patients in the G1 group continued the kinetic program at home, at the rate of three sessions per week, each with a duration of 30 min, which allowed them to obtain better results than the patients in the G2 group. These differences allowed us to make some assessments: 

1. The pain was relieved through electroanalgesic therapy, alongside physical therapy, aligning with findings from specialized studies. Controlled studies have indicated the analgesic effect of electrotherapy between 25–90%, with no side effects and few contraindications. One of the analgesic procedures for electrotherapy is TENS, which can be an alternative to NSAIDs, which have side effects. Studies showed a reduction in pain and a decrease in the amount of NSAIDs used [86,87].

The results of a 2018 study [88] showed that men are more likely to get sick, with pain being a predominant symptom. 

Other studies [89,90,91] showed a predominance of pain among women, an aspect that must be addressed to determine whether it is due to the preponderance of gender or an imbalance in the population in favor of more women than men. However, the results of physiotherapy should not be neglected. For this purpose, there is research that shows that the use of physical therapy as an active therapy has better results than the passive modality represented by electrotherapy: ultrasound, TENS [92]. 

2. Lumbar spine disability and mobility improved primarily due to the administered physical therapy, as indicated in several studies [93,94,95,96]. The disability and mobility of the lumbar spine improved primarily due to the implemented physical therapy, as supported by findings in various studies. The Bernadellis study [97] highlighted that incorporating regular exercise at home or at work presents an opportunity to enhance health and diminish disability.

Studies [98,99] show that reducing the Oswestry Index score represents a decrease in pain intensity, with an increase in ROM and functional activities.

3. Electrotherapy had an impact on heart rate and blood pressure, with physical therapy showing a particularly significant influence.

The changes in blood pressure and heart rate were greater in the G1 group. Studies show that electrotherapy can influence the circulatory system to increase local vasodilation and hyperemia at the application site, a phenomenon that can have an effect on local and general metabolism. Practicing therapeutic, controlled physical exercise, both during the treatment period and in the period leading up to the control, allowed the adaptation of these parameters to efforts over time; therefore, no high blood pressure or heart rate values were recorded.

Heart rate variability (HRV) reflects the modulation of the autonomic nervous system under various conditions, such as sleep, body position, psychological state, and physical activity [100,101]. Efferent autonomic control of the sinus node, particularly with respect to timing, is a major factor in determining HRV. For example, Catai’s study [102] looked at the impact of aerobic exercise on HRV and cardiorespiratory responses, finding that heart rate is influenced by age. Other research [103,104,105] indicates that maximal aerobic capacity peaks around age 30 and declines thereafter, associated with a decrease in parasympathetic tone.

Subsequent studies in subjects over 60 years of age have shown that sustained exercise for 6–12 months can improve parasympathetic activity, increase HRV, and reduce nocturnal heart rate [106,107]. It is important to note that the balance between sympathetic and parasympathetic tone, with a predominance of the latter, influences resting HR. For example, increased vagal tone is thought to be the primary mechanism in exercise-induced bradycardia over a period of 6 months to 1 year [108]. However, there are also studies [101,109,110] that failed to reveal significant differences between subjects who exercised and those who did not for a longer period.

The results of Catai’s study are consistent with other research [101,109,110], which suggested that there were no significant differences between young adults [111] and older adults [112] following a program of physical exercise. 

Fohr’s study [113] also highlights the dynamics of cardiac autonomic activity during the day, influenced by factors such as stress, relaxation, sleep and physical activity. Other research [114,115] supports the association of physical activity with an increase in heart rate variability, indicating that long-term physical activity can improve physical fitness and health. Assessing the interconnection of the central nervous system and the heart through HRV, according to the methods presented by Thayer [116,117], has become a noninvasive way to assess sinus node modulation.

According to the study of Appelhans [118], an increased variability of cardiac activity at rest may indicate a context of highly adaptive emotional responses. In contrast, low HRV is associated with cardiovascular disease, mood disorders, and other pathologies. Furthermore, research such as that of Gockel [119], Cohen [120], and Moustoufi [121] has shown decreased HR in low back pain, fibromyalgia, and neck pain, indicating that pain may alter sympatho-vagal balance, and heart rate may serve as a marker for physical and psychological disorders, especially in the case of chronic pain.

Bandeira’s study [122] focused on identifying resting HRV in patients with LBP. The results suggested an increase in sympathetic outflow, as measured by HR values, in these patients, indicating decreased vagal tone. In addition, Pigozzi’s study [123] revealed an increase in sympathetic modulation of sinus node activity in individuals who performed regular exercise for 5 weeks, which may coexist with signs of reduced vagal modulation activity.

Perrini’s study [124] demonstrated that regular physical activity can influence vagal and cardiac sympathetic activities. This impact can be attributed to the increase in respiratory rate, determined by the practice of physical exercise, which, in turn, has a direct effect on heart rate modulation. Sandercock’s [125] meta-analysis based on 13 studies confirmed the effects of exercise training on HR and explored the association between HRV and cardiac vagal modulation.

Hottenrott’s study [126] emphasized the use of HRV in the assessment of short- and long-term autonomic changes in endurance exercise, whether it is practiced for leisure or in high-performance activities. HRV is a significant marker in the diagnosis of strain and overstrain. Assessed in healthy individuals and those with cardiovascular disease, HRV shows significant improvements after regular exercise. Changes accompanied by reductions in HR at rest and during exercise reflect an increase in autonomic efferent activity and a change in vagal modulation of heart rate.

Tuomainen’s study [127] also highlighted changes in HRV in a 6-month randomized trial, highlighting the beneficial effects on cardiac autonomic nervous function, which is a relevant predictor of cardiovascular morbidity.

In the context of physiological pain assessments, parameters such as HR, HRV and respiratory rate (RR) have become topics of interest. HRV, a parameter proposed by Forte [128], is considered an indicator of the flexibility and adaptive regulation of the nervous system, providing a homeostatic response to factors such as stress, environment and physical activity.

HRV is considered an optimal variable for measuring the sensitivity of the autonomic nervous system (ANS) to nociceptive stimulation, including in pain situations. Since HRV reflects the body’s ability to adapt to changes in circumstances, including in the presence of pain [129], it becomes an important indicator for evaluating the modulation of the autonomic nervous system, including both sympathetic and parasympathetic components [128].

The strong interaction between the nociceptive nervous system and the autonomic nervous system, in both the sympathetic and parasympathetic branches, has been highlighted in studies such as those of Benarroch and Pham [130,131].

Studies, such as those of Younes [132], Zhang [133] and Bandeira [122], have proposed the inclusion of HRV as a relevant parameter in the assessment of pain, benefits of treatments and sympathetic-parasympathetic balance associated with LBP.

Pain can also be considered a stressful situation for our body, and HRV is known to be altered in stressful situations, according to studies such as those of Attar [134] and Schneider [135].

The recent study by Jacinto et al. [136] evaluated the effects of two 24-week exercise programs on indices of health and fitness in institutionalized individuals with intellectual and developmental disabilities. Study findings revealed significant differences in cardiovascular response, with significant changes in HR observed at midterm assessments, while SBP and DBP variables did not show significant differences between groups.

There were significant differences between the initial and final values for SDP and DBP, as were recorded in our study.

There is consistent evidence indicating that regular physical exercise significantly influences cardiovascular parameters. Boer’s study [137] and Reimesr’s [138] meta-analysis found a significant decrease in SBP after the final intervention, suggesting that exercise can reduce SBP. There is also a negative association between participant age and decreased HR, and a high resting HR is associated with increased all-cause mortality [139].

This aspect can be explained by endothelial dysfunction, reduction of arterial distensibility, increased stress on the arterial wall and increased pulse wave speed [140].

Cardiac autonomic control is influenced not only by physical exercise, but also by the time of day the physical activity takes place, according to studies by Waninge [141] and Sandercock [125]. Exercise appears to exert an antiarrhythmic effect, inducing resting bradycardia and increased cardiac vagal modulation [125].

Exercise modulates cardiac autonomic control by decreasing sympathetic influence and increasing vagal tone. This shift toward greater vagal modulation can positively affect the prognosis of individuals with a variety of morbidities [142].

An interesting aspect is the resetting of the arterial baroreflex at the onset of physical exercise, indicated by the study of Michael et al. [143]. This reset involves a physiological change in cardiac output and increased sympatho-vagal activity, but is short-lived [144]. Studies such as Berry’s in 2020 [145] indicate post-exercise parasympathetic reactivation evident in HR dynamics 5 min after exercise.

LBP can cause disability [146] and inability to perform daily activities, including workplace absenteeism [147], representing a burden on the economic and health systems [148].

4. The oxygen saturation of the blood increased initially in both groups, later only in the G1 group, due to the ability to adapt to the physical effort, which is in agreement with specialized studies.

5. The quality of life has improved significantly through the regular application of therapeutic physical exercises. The results obtained regarding the relationship between physical activity and LBP are consistent with those reported in studies [91,92]. Other studies [149,150] evaluated the action of physical activity in reducing pain, improving physical function and physical quality. The effects of regular physical activity of moderate intensity reduce the risk of morbidity and mortality [151,152,153], also having a prophylactic role for cardiovascular diseases, diabetes, osteoporosis, depression and some musculoskeletal disorders, including LBP [154]. Exercise improves the quality of life of patients with LBP. Therefore, physical activity is recommended in clinical guidelines [154,155,156]. In the case of chronic low back pain, it can be seen that the physical therapy exercise approach remains a first-line treatment and should be used systematically [155]. On the other hand, the positive responses of the nervous system to pain are represented by the increase in RR, HR, BP, and vegetative phenomena (sweating, anxiety, restlessness) [157].

Physical therapy is an essential element in the management of LBP, addressing both physical and emotional aspects of the patient. Thus, physical therapy aims to increase muscle strength, with the objectives of reducing pain, improving function and accelerating the patient’s recovery with the return to usual activities. In addition to the physical benefits of exercise, there are also emotional and psychological benefits, which can decrease pain and improve functional capacity. The management of LBP is complex and costly, with social, economic and medical implications. Disability caused by LBP is a major problem behind workplace absenteeism. Also, the prevalence of LBP in the active population (25–64 years) is increasing [89,158]. LBP is one of the most prevalent and disabling conditions, and its economic burden is enormous. For many patients, especially young ones, the symptoms tend to become chronic; therefore, the treatment should be a biopsychosocial approach, based on evidence and cost-effectiveness, especially in the early stages, as the specialized literature shows [159]. LBP has became a growing problem worldwide [61,160]. That is why specialists in medical recovery should offer patients treatment but also education regarding the causes, mechanisms and prognosis of this disabling condition, promoting the benefits of therapeutic exercises and electric modalities [161].

There are also limitations of the study, namely the small number of patients and the relatively short period of evaluation of changes in physiological parameters in this condition. 

We can include as limitations the presence of variables that were not taken into account in the design, bias in the selection or the interpretation of the data, measurement errors and subjectivity of the recordings.

A study from 2020 [162] highlights that depending on the type of work practiced, it is necessary that the exercise program be adapted to the professional activity, mentioning that future studies and on numerous groups are needed to confirm the results of this study.

Discussion: Integrating Cardiovascular Parameters in Rehabilitation Studies

Our study, focusing on the impact of tailored therapeutic exercises, including elements of electrotherapy, opens the possibility of exploring the effects of physical therapy on cardiovascular parameters. The literature provides valuable insights into the interconnectedness of musculoskeletal and cardiovascular health, highlighting the potential benefits of comprehensive rehabilitation programs.

1. Exercise and Cardiovascular Health: Numerous studies have emphasized the positive influence of exercise on cardiovascular parameters. Regular physical activity has been associated with improved HR, reduced risk of cardiovascular diseases, and enhanced cardiovascular function [163]. In our study, the prescribed therapeutic exercises may have contributed not only to musculoskeletal improvements but also to cardiovascular well-being.

In order to promote and maintain health, it is recommended to practice physical exercise of moderate intensity (accelerates the heart rate), 30 min/day, five times a week, or of increased intensity (fast breathing and a substantial increase in the heart rate) for at least 20 min/day, three times a week.

2. Electrotherapy and Circulatory Benefits: Electrotherapy, as an adjunct to exercise, has been explored for its potential cardiovascular effects. For instance, TENS has been suggested to enhance blood circulation by promoting vasodilation and reducing sympathetic nervous system activity [164]. This suggests that the electrotherapy component in our study may have played a role in influencing cardiovascular parameters.

3. Synergistic Effects of Exercise and Electrotherapy: Combining exercise and electrotherapy may result in synergistic effects on cardiovascular health. A study [165] demonstrated that a combination of therapeutic exercise and electrical muscle stimulation positively impacted endothelial function, a key indicator of cardiovascular health. This supports the notion that our integrated approach could have multifaceted benefits, addressing both musculoskeletal and cardiovascular aspects.

Moreover, high-intensity exercise has been found to reduce pain, disability rate, and psychological stress in PML patients [90,166,167].

4. Considerations for Future Research: While our study provides a foundation, further research is warranted to delve into the specific mechanisms underlying the cardiovascular effects of combined physical therapy interventions. Longitudinal studies with comprehensive cardiovascular assessments, such as of blood pressure, heart rate variability, and endothelial function, could provide a more nuanced understanding of the holistic impact of tailored therapeutic exercises.

Exercise may also improve cardiovascular function through adaptations of the heart and vascular system [168,169,170]. Regular exercise reduces resting HR, BP, and atherogenic markers and increases physiological cardiac hypertrophy [171,172,173,174]. Exercise improves myocardial perfusion and increases high-density lipoprotein (HDL) cholesterol, all of which reduce stress on the heart and improve cardiovascular function in healthy and diseased individuals [173,175,176,177]. There is increasing interest in exercise-based therapies due to the beneficial effects of exercise on cardiovascular health and the potential mechanisms by which they occur.

In conclusion, our study not only contributes to the rehabilitation literature but also opens avenues for investigating the cardiovascular implications of physical therapy interventions. By acknowledging the interconnectedness of musculoskeletal and cardiovascular systems, future research can advance our understanding and refine rehabilitation strategies for optimal patient outcomes.

## 5. Conclusions

This study examined the role of electrotherapy combined with exercise therapy in the treatment of low back pain in young adults. The patients who participated in the study felt that the values of some cardiovascular parameters were influenced during the use of electrotherapy and individualized kinesitherapy (related to the clinical-functional status of each patient), values that can later return to their initial levels, especially through training and by adapting cardiac function to physical effort. Decreased pain and disability and increased flexibility of the lumbar spine were also noted, particularly through the application of therapeutic exercises. 

The study is a starting point for future research on the possible influence of cardiovascular parameters when combined with electrotherapy and therapeutic exercise in the treatment of back pain. 

Depending on the clinical-functional status of each patient, physical therapy can accelerate the heart rate and increase blood pressure and oxygen saturation of arterial blood, values that can later return to their initial levels, especially through training. 

The use of complex and individualized treatment that includes electrotherapy and kinesitherapy causes a decrease in pain syndrome and disability, an increase in the flexibility of the lumbar spine, reducing the values of the finger-to-ground index, and adaptation of cardiac function to effort.

## Figures and Tables

**Table 1 healthcare-12-00853-t001:** Criteria for inclusion/exclusion.

Inclusion Criteria	Exclusion Criteria
Age 20–40 Pain in the lumbosacral region for at least 3 months No acute/inflammatory conditions or associated comorbidities	Age < 20 and over 40 Surgery on the lumbosacral spine under 6 months Trauma Neoplasms Pregnancy Decompensated diseases Mental disorders Non-cooperating patients

**Table 2 healthcare-12-00853-t002:** Clinical and phenotypic characteristics of study groups.

Characteristics	TOTAL—175		G1—86		G2—89	
	N	%	N	%	N	%
Age, years						
20–25	41	23.43	15	17.45	26	29.21
26–30	36	20.57	20	23.25	16	17.98
31–35	59	33.71	28	32.55	31	34.83
36–40	39	22.29	23	26.75	16	17.98
Gender						
Male	86	49.14	42	48.84	44	49.44
Female	89	50.86	44	51.16	45	50.56
Geographical area						
Urban	115	65.71	55	63.95	60	67.42
Rural	60	34.29	31	36.05	29	32.58
Occupation of participant						
Student	19	10.85	10	11.62	9	10.12
Teacher	35	20	18	20.93	17	19.11
Marketing field	22	12.57	10	11.62	12	13.48
Driver	50	28.59	24	27.91	26	29.21
Daily laborer	29	16.57	15	17.45	14	15.73
Other occupations	20	11.42	9	10.46	11	12.35
Work absenteeism						
Teacher	4	2.28	2	2.32	2	2.24
Marketing field	4	2.28	1	1.16	3	3.37
Driver	11	6.28	5	5.81	6	6.74
Daily laborer	6	3.42	2	2.32	4	4.49
Other occupations	3	1.71	1	1.16	2	2.24

**Table 3 healthcare-12-00853-t003:** T-student assay values in groups G1 and G2.

	G1 Group			G2 Group		
	T1–T2	T2–T3	T1–T3	T1–T2	T2–T3	T1–T3
VAS	0.020167	0.030951	0.009173	0.015006	0.029831	0.007017
LBP-M	0.047285	0.025382	0.015833	0.048983	0.021924	0.014757
ODI	0.029079	0.044103	0.016417	0.008673	0.013155	0.002053
FFDI	0.028667	0.049014	0.018713	0.001746	0.001207	0.000004
SBP	0.001967	0.000003	0.000009	0.001813	0.000009	0.000005
DBP	0.003244	0.000522	0.000001	0.001859	0.000391	0.000004
HR	0.003145	0.000679	0.000002	0.000124	0.001326	0.000007
SAT O_2_	0.000001	0.000005	0.000008	0.000001	0.000004	0.000005

**Table 4 healthcare-12-00853-t004:** T-student assay values for subgroups of women from groups G1 and G2.

	G1 Group			G2 Group		
	T1–T2	T2–T3	T1–T3	T1–T2	T2–T3	T1–T3
VAS	0.020872	0.031579	0.009669	0.014265	0.028998	0.006552
LBP-M	0.048647	0.024483	0.015871	0.048937	0.021294	0.014478
ODI	0.025892	0.038998	0.013525	0.008354	0.012674	0.001922
FFDI	0.030301	0.046999	0.018923	0.001796	0.001551	0.000005
SBP	0.001884	0.000005	0.000008	0.001592	0.000007	0.000004
DBP	0.002782	0.000361	0.000001	0.001965	0.000451	0.000005
HR	0.001736	0.000429	0.000007	0.000132	0.001292	0.000007
SAT O_2_	0.000001	0.000009	0.000007	0.000002	0.000003	0.000005

**Table 5 healthcare-12-00853-t005:** T-student assay values for subgroups of male patients from groups G1 and G2.

	G1 Group			G2 Group		
	T1–T2	T2–T3	T1–T3	T1–T2	T2–T3	T1–T3
VAS	0.019183	0.029801	0.008463	0.015394	0.029132	0.000702
LBP-M	0.046131	0.026393	0.015888	0.048802	0.022835	0.015102
ODI	0.032151	0.048821	0.020302	0.009249	0.014191	0.002327
FFDI	0.027047	0.046686	0.017227	0.001717	0.000927	0.000003
SBP	0.002064	0.000003	0.000001	0.001935	0.000132	0.000006
DBP	0.003901	0.000749	0.000002	0.001814	0.000382	0.000004
HR	0.005053	0.001011	0.000006	0.000108	0.002275	0.000008
SAT O_2_	0.000001	0.000002	0.000009	0.000001	0.000004	0.000005

## Data Availability

Data are contained within the article.

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
