# Peer review of "Lumbar Paravertebral Muscle Pain Management Using Kinesitherapy and Electrotherapeutic Modalities"

_healthcare, 2024, doi:10.3390/healthcare12080853_

Round 1

Reviewer 1 Report (Previous Reviewer 2)

Comments and Suggestions for Authors

Dear authors, these are my comments and suggestions.

I see that the abstract has been divided into the different sections. But please remove the numbers.

I believe that the introduction provided contains too much information on the various non-pharmacological treatment strategies for low back pain such as exercise therapy, lumbar stabilisation exercise programmes, therapeutic ultrasound, transcutaneous electrical nerve stimulation and low level laser therapy. However, it lacks a specific focus on the aims and hypotheses of the presented study.

It should be summarised, and should aim more at summarising previous research related to the efficacy of these strategies in the management of low back pain. Sometimes it feels like reading a book and not the introduction to a scientific article. It would need a clearer transition between paragraphs and linking of the different paragraphs.

Methods

-When I reviewed this study in January, it said it was for two years and here it says 2 months? I commented the first time I read the article, that it would be necessary to justify why the length of the trial was so long.

-  It is important to specify that this is a prospective longitudinal study to provide a more precise description of the research design used. I proposed this in January and I still do not see it specified.

-The section "Statistical analysis" is rather terse and lacks details on the specific methods used for data analysis. Details on the statistical tests used are lacking. It would be useful to provide information on the specific tests used, such as Student's t-test, ANOVA, chi-square test, etc., and to justify it. It should be stated whether adjustments were made to correct potential errors.

Mentioning that Excel was used and saying it in the first person (it is better to do it impersonally) is unscientific, as Excel is not the standard software used in research. It would be better to use a widely accepted statistical software, such as SPSS, R.....

The effect size should also be included along with the p-values to provide a fuller understanding of the practical significance of the results found.

-In lines 257 and 272, the justification for the applied physiotherapy is discussed; however, there are no references provided to support this assertion

-I've noticed that the suggestion that I have mentioned in January, regarding exercises at home were made.

Results

-  In order to reduce the length of the results section, it would be advisable to leave the most important tables and put those of lesser value in annexes. Highlight the most significant results in the main text, and mention that additional data, tables, and statistical details can be found in the appendix.

Discussion

- Lines 819-828. Regarding the limitations of the study, I have noticed that the authors have increased them. However, the reference to a 2020 study, mentioned in relation to adapting the exercise program to professional activity, appears to be unrelated to the specific limitations of this particular study. It would be helpful to provide a clearer justification of how this reference supports the specific limitations mentioned earlier, such as the limited number of patients and the short evaluation period.

Additionally, the term "highlights" in this context seems to be misused. It would be more appropriate to use words like "findings" or "results" to refer to the results of the referenced study.

The discussion needs to be revisited. It should be structured more effectively to enhance clarity. There are various enumerations, and some paragraphs lack coherence with the preceding ones. For example Line 829

Author Response

AUTHORS’ RESPONSE TO THE COMMENTS FROM THE REVIEWER 1

We appreciate your time and efforts for generating these valuable comments. The point-by-point responses can be seen in the WORD file we uploaded and in our revised manuscript.

Comments and Suggestions for Authors

Dear authors, these are my comments and suggestions.

I see that the abstract has been divided into the different sections. But please remove the numbers.

Response: Thank you, I removed the numbers from the abstract.

I believe that the introduction provided contains too much information on the various non-pharmacological treatment strategies for low back pain such as exercise therapy, lumbar stabilisation exercise programmes, therapeutic ultrasound, transcutaneous electrical nerve stimulation and low level laser therapy. However, it lacks a specific focus on the aims and hypotheses of the presented study.

Response: We reviewed the text and tried to focus on the requested ones.

It should be summarised, and should aim more at summarising previous research related to the efficacy of these strategies in the management of low back pain. Sometimes it feels like reading a book and not the introduction to a scientific article. It would need a clearer transition between paragraphs and linking of the different paragraphs.

Response: We reviewed the text taking into account your indications.

Methods

-When I reviewed this study in January, it said it was for two years and here it says 2 months? I commented the first time I read the article, that it would be necessary to justify why the length of the trial was so long.

Response: There was an error in the translation of the text. The study period was 6 months

-  It is important to specify that this is a prospective longitudinal study to provide a more precise description of the research design used. I proposed this in January and I still do not see it specified.

Response: Yes, I added this to the text.

-The section "Statistical analysis" is rather terse and lacks details on the specific methods used for data analysis. Details on the statistical tests used are lacking. It would be useful to provide information on the specific tests used, such as Student's t-test, ANOVA, chi-square test, etc., and to justify it. It should be stated whether adjustments were made to correct potential errors.

Response: I made the requested changes

Mentioning that Excel was used and saying it in the first person (it is better to do it impersonally) is unscientific, as Excel is not the standard software used in research. It would be better to use a widely accepted statistical software, such as SPSS, R.....

Response: I made the changes in the text.

The effect size should also be included along with the p-values to provide a fuller understanding

Response: I made the necessary statements in the text

-In lines 257 and 272, the justification for the applied physiotherapy is discussed; however, there are no references provided to support this assertion

Response: In the text, in rows 137-176 I made the specifications regarding the physiotherapy applied, with references from literature of specialty.

-I've noticed that the suggestion that I have mentioned in January, regarding exercises at home were made.

Response: Yes, I applied your suggestion.

Results

-  In order to reduce the length of the results section, it would be advisable to leave the most important tables and put those of lesser value in annexes. Highlight the most significant results in the main text, and mention that additional data, tables, and statistical details can be found in the appendix.

Response: I took your suggestion into account and made the necessary changes. We moved the tables in the Additional Materials section

Discussion

- Lines 819-828. Regarding the limitations of the study, I have noticed that the authors have increased them. However, the reference to a 2020 study, mentioned in relation to adapting the exercise program to professional activity, appears to be unrelated to the specific limitations of this particular study. It would be helpful to provide a clearer justification of how this reference supports the specific limitations mentioned earlier, such as the limited number of patients and the short evaluation period.

Response: We have removed the study you are referring to

Additionally, the term "highlights" in this context seems to be misused. It would be more appropriate to use words like "findings" or "results" to refer to the results of the referenced study.

Response: I made the correction it in the text

The discussion needs to be revisited. It should be structured more effectively to enhance clarity. There are various enumerations, and some paragraphs lack coherence with the preceding ones. For example Line 829

Response: We reviewed the Discussion chapter.

I revised the overall grammar and syntax errors.

Reviewer 2 Report (Previous Reviewer 1)

Comments and Suggestions for Authors

Introduction: 

Line 88 sentence should be modified to ( the effects of PA not only improve cardiovascular capacity but also minimize MSD,cancers and other ...

Line 92 change to sleep disturbance or poor sleep quality ,remove it from end of sentences .Also replace disability by inability to perform ADLs . All of the these aggrivate intesity of MS Pain.

Line 101 replace it to theraputic exercise the same in line 109 .check the whole text just use this term .line 110 the benift should be wriiten as the following reduction in pain ,provement in functional performance. 

Line 132 remove *man ......thanks to myocardium doesn't have meaning .

Methods:

Line 191 change content tothe treatment procedure .

Line 213 change painful conditions to Patients with LBP.

Need to elaborate more about the questions and information that you will gain using LBP Module scale &Owstery index.

Shift semtence of measuring BP to separate paragraph .

Need to discuss duration of treatment ,as it is not enough only to write the type of electromodalities that were used you have to talk about the time of application of each device ,the frequency of sessions and the whole duration of treatment.

Line 272 need to add how many sessions/week .the total sessions. For exsmple  (3sessions/week for 10 weeks .

Line 286 revise the sentence (back school??????.

Lone 296 how you can asses please use past tense .

Did you use William program as your writing is not reflecting that you isrd it however this onformation cpuld be added in ontroduction. In methods you have to write how you applied tge exercise .

Table 2 and participants charchtristics should be shifted to results.

Line 350 talking about ergonomecis by this way could be added in introduction  however on methods you should mention if you did ergonomic assrsment for Patients before engaging them in treatment protocol.

Results :

Needore narritive analysis for tesults .Tables are not enough

Discussion : remove thanks to...

.663 how both lumber disability and mobility  improved at the same time please revise the whole section

Line 671 do you mean but theraputic exercise? Instead of PT. Use the padt tense allover the article either when talking about your methods or discussing findings of previous studies .please check the whole draft.

Make sure that all abbreviations should be written fully first time.

Limitation should be written at the end of argicle with adding recommendations for future resesrch. Please rearrange discussion section.

Why you write conclusion twice? Correct the whole manscuript. 

Please revise the whole manuscript and midify its style.

Comments on the Quality of English Language

Please revise the whole article 

Author Response

AUTHORS’ RESPONSE TO THE COMMENTS FROM THE REVIEWER 1

We appreciate your time and efforts for generating these valuable comments. The point-by-point responses can be seen in the WORD file we uploaded and in our revised manuscript.

Comments and Suggestions for Authors

Introduction: 

Line 88 sentence should be modified to (the effects of PA not only improve cardiovascular capacity but also minimize MSD,cancers and other ...

Response:I corrected it in the text

Line 92 change to sleep disturbance or poor sleep quality,remove it from end of sentences .Also replace disability by inability to perform ADLs . All of the these aggrivate intesity of MS Pain.

Response:I made the correction in the text

Line 101 replace it to therapeutic exercise the same in line 109 check the whole text just use this term .line 110 the benift should be wriiten as the following reduction in pain ,provement in functional performance. 

Response:I made the replacement in the text

Line 132 remove *man ......thanks to myocardium doesn't have meaning.

Response:I corrected

Methods:

Line 191 change content tothe treatment procedure .

Response:I corrected it in the text

Line 213 change painful conditions to Patients with LBP.

Response:I corrected it in the text

Need to elaborate more about the questions and information that you will gain using LBP Module scale &Owstery index.

Response:I have completed the explanations in the text

Shift semtence of measuring BP to separate paragraph .

Response:I corrected it in the text

Need to discuss duration of treatment ,as it is not enough only to write the type of electromodalities  that were used you have to talk about the time of application of each device, the frequency of sessions and the whole duration of treatment.

Response:In the text in the rows 196-198 we specified that the duration of treatment is 15 days. Physiotherapy and kinesiotherapy applications were carried out daily during this period.

Line 272 need to add how many sessions/week . the total sessions. For exsmple  (3sessions/week for 10 weeks .

Response:The duration of treatment is 15 days, it is specified in row 197 and in row 269 it is shown duration per session of 45 minutes

Line 286 revise the sentence (back school??????.

Response:I specified in the text

Lone 296 how you can asses please use past tense .

Response:I corrected it in the text

Did you use William program as your writing is not reflecting that you isrd it however this onformation cpuld be added in ontroduction. In methods you have to write how you applied tge exercise .

Response:Elements of the Williams exercises are shown in rows 270-284

Table 2 and participants charchtristics should be shifted to results.

Response:The other reviewers wanted the tables starting with the number 3 to switch to additional material. We'll do what you think is best.

Line 350 talking about ergonomecis by this way could be added in introduction  however on methods you should mention if you did ergonomic assrsment for Patients before engaging them in treatment protocol.

Response:I corrected it in the text

Results :

Needore narritive analysis for tesults .Tables are not enough

Response:I moved the tables to the extra material

Discussion : remove thanks to...

Response:I corrected it in the text

.663 how both lumber disability and mobility  improved at the same time please revise the whole section

Response:I reviewed those phrases

Line 671 do you mean but theraputic exercise? Instead of PT. Use the padt tense allover the article either when talking about your methods or discussing findings of previous studies .please check the whole draft.

Response:I made the change in the text

Make sure that all abbreviations should be written fully first time.

Response:I checked the text

Limitation should be written at the end of argicle with adding recommendations for future resesrch. Please rearrange discussion section.

Response:I corrected the discussion section

Why you write conclusion twice? Correct the whole manscuript. 

Response:I made the requested corrections

Please revise the whole manuscript and midify its style.

Response:We reviewed what you requested.

Comentarii despre Calitatea Limbii Engleze

Response:I have made the corrections indicated by you

I revised the overall grammar and syntax errors

Reviewer 3 Report (New Reviewer)

Comments and Suggestions for Authors

Thank you for a very interesting manuscript. I have a few comments.

1. Abstract: There you mention that low back pain ....in adolescents and adults as well. I thought that this manuscript included adolescents specifically but then I realised it was young adults from 20-40 years of age. LBP occurs in people in all ages so I found this confusing.  You also mention in the abstract that the study was carried out over a period of 2 years but in the method chapter there is 2 months. Which time frame is correct? I found the conclusion in the abstract could be improved and be more like the one in the manuscript.

The introduction and the methods are very well described but I found it difficult to read the results. There are so many tables and in the text there you describe reduction and decrease in % which I can't see in the tables. Would it be possible to decrease the numbers of tables and put in the % and p-value in the tables? I believe that there are some typing errors in line 444 (at time of T3-should be T2?) and again in line 502 (T3 should be T2?) and line 563 (T3 should be T2). In table 16 there are G1 and G2 and L1 and L2 groups. Is it supposed to be like that? I found it difficult to read the text after table 16 because the table use VAS, LBP, ODI etc but in the text there is e.g. pain relief, quality of life, blood oxygen saturation etc Is it possible to change that? You said in the discussion that the patients continued the program at home. How do you know that?

Author Response

AUTHORS’ RESPONSE TO THE COMMENTS FROM THE REVIEWER 1

We appreciate your time and efforts for generating these valuable comments. The point-by-point responses can be seen in the WORD file we uploaded and in our revised manuscript.

Comments and Suggestions for Authors

Thank you for a very interesting manuscript. I have a few comments.

Abstract: There you mention that low back pain ....in adolescents and adults as well. I thought that this manuscript included adolescents specifically but then I realised it was young adults from 20-40 years of age. LBP occurs in people in all ages so I found this confusing.  You also mention in the abstract that the study was carried out over a period of 2 years but in the method chapter there is 2 months. Which time frame is correct? I found the conclusion in the abstract could be improved and be more like the one in the manuscript.

Response: Thank you, I corrected it in the summary. The study period was 6 months

The introduction and the methods are very well described but I found it difficult to read the results. There are so many tables and in the text there you describe reduction and decrease in % which I can't see in the tables. Would it be possible to decrease the numbers of tables and put in the % and p-value in the tables? I believe that there are some typing errors in line 444 (at time of T3-should be T2?) and again in line 502 (T3 should be T2?) and line 563 (T3 should be T2). In table 16 there are G1 and G2 and L1 and L2 groups. Is it supposed to be like that? I found it difficult to read the text after table 16 because the table use VAS, LBP, ODI etc but in the text there is e.g. pain relief, quality of life, blood oxygen saturation etc Is it possible to change that? You said in the discussion that the patients continued the program at home. How do you know that?

Response: Thank you, we corrected the errors in rows 444, 502, 563. It is T2 and not T3.

In Table 16 we corrected and instead of L1 group and L2 group is G1 group and G2 group.

Patients continued at home the School Back program. We specified in the text in rows 282-284 that the study participants received verbal and written instructions regarding the exercises to be performed at home and the importance of their execution.

They had a journal of activity that they presented.

Round 2

Reviewer 2 Report (Previous Reviewer 1)

Comments and Suggestions for Authors

All recommendations have been addressed 

Best regards.

This manuscript is a resubmission of an earlier submission. The following is a list of the peer review reports and author responses from that submission.

Round 1

Reviewer 1 Report

Comments and Suggestions for Authors

‘’ Lumbar paravertebral muscles pain management using physical exercises and electric therapeutical modalities’’

Thanks for the study. I would like to make a some comments, which I hope will serve to improve the quality of the manuscript

I suggest that the title could be improved to be more reflective of the study .could be Kinetic and electotherapy

Abstract : should be structured according to MDPI policy ( background , aim, …etc. )

You mentioned that :The purpose of the paper is to highlight: the physiological and functional changes in young adults with painful conditions of the lumbar spine, after using therapeutic physical exercises. However in the title you mentioned both exercise and electrotherapy . This is confusing

The term therapeutical currents should  be changed to be electro therapy or TENS or the modalities you used .

In results : physical activity should be changed as it may be determined as  walking or any type of activities . I thought you were talking about therapeutic exs. Even conclusion is not matching the title.

Introduction :

You spent too much describing back pain, However you need more articles for back pain management with different protocols .  

Line 115- 133 not supported with evidence from Literature .

Just use one term related to your intervention was it exercise or activity  or kinetic therapy ??? KT is good. Use it all over the article.

Need to show significance of study and what it may add to the field .

Methods :

Regarding pain How did you select participants .Pain only is not enough, Please consider the cause of pain ,was the pain related to specific diagnosis as you mentioned that their was imaging , What were the job of participants, where they office worker for example  ???

Thin to chest ???????, Do you mean chin please correct.

Remove this highlighted and consider revision of whole manscuript

(submission, please state that they will be provided during review. They must be pro- 251

vided prior to publication. 252

Interventionary studies involving animals or humans, and other studies that require 253

ethical approval, must list the authority that provided approval and the corresponding 254

ethical approval code

Presentation of the group :. 

This should be shifted to results not in methods. Need to add theie marital and employment status  

Discussion:

Lumbar spine disability and mobility improved mainly due to applied physical  therapy  .Do you mean in both groups ( electro and kinetic) ? Please specify .I see that you are confusing between  PT and exercise .PT consist of both so remove PT and replace it by KE.

Please write all limitations in the study .

Please revise reference list ,some typo issues .

Comments on the Quality of English Language

Article should be revised and edited.

Author Response

Hello,

We appreciate your time and efforts for generating these valuable comments.

We are attaching a document with our answers.

Thank you

Reviewer 2 Report

Comments and Suggestions for Authors

First of all, thank you for allowing me to review your manuscript.

This paper examines the physiological and functional changes in young adults with lumbar spine pain after undergoing therapeutic physical exercises in a two-year longitudinal study. The results reveal statistically significant improvements across various parameters, underscoring the importance of exercise in alleviating pain and enhancing mobility in the lumbosacral spine.

After reading in depth the manuscript, I would like to make some comments and ask the authors several questions about.

Introduction

- It is necessary to go through the whole text because there are paragraphs in red.

-In my view the authors should clarify and highlight from the outset the main objective of the study, which is to investigate the influence of electrotherapy combined with exercise on pain management in the lumbar paravertebral muscles.

- I recommend reducing the length of the introduction. Details that are not directly related to the main objective of the study should be removed to maintain relevance and capture the reader's attention more efficiently.

-There should be a brief contextualisation about electrotherapy in the introduction. Background information related to electrotherapy and its application in the management of low back pain should be briefly incorporated to establish the rationale and relevance of the research.

- Paragraphs would need to be organised in a way that leads naturally to the main objective of the study, avoiding redundant or distracting information.

Material and methods

- It is mentioned that it is a 2-year longitudinal study, but the exact study design should be specified (e.g. prospective, retrospective).

- Given that the study was conducted over a 2-year period, you might suggest that the authors provide a brief justification for the chosen duration. Was this duration necessary to capture changes over time?

- It is mentioned that after treatment, patients continued with therapeutic exercises at home. It would be useful to include how this follow-up was monitored, whether specific guidelines were provided and whether regular assessments were made.

Results

-  In order to reduce the length of the results section, it would be advisable to leave the most important tables and put those of lesser value in annexes. Highlight the most significant results in the main text, and mention that additional data, tables, and statistical details can be found in the appendix.

Discussion

- The section on limitations is very brief. Possible biases, methodological difficulties or any other constraints that may affect the interpretation of the results could be reflected upon.

References

- Review reference 25 which is underlined.

-  Review reference 78, which is in red.

Author Response

(The authors gave the same response as above.)

Reviewer 3 Report

Comments and Suggestions for Authors

See attached PDF.

Comments on the Quality of English Language

Check all english and grammar.

Author Response

(The authors gave the same response as above.)

Round 2

Reviewer 1 Report

Comments and Suggestions for Authors

Thanks for modification .

Please consider the following to enhance quality of article:

In results : you mentioned that pain was improved 50% ,please changed to relief not improve.

In methods you described the occupation of Participants, however you didn't mention how each cluster responded to diffrent interventions . 

You mention that you will gave them home program ,but you didn’t confirm how ergonomics may impact them .

Can you ellaborate about if Participants were in sick leave during the experiment or they were attending their jobs??

You mentioned that sustolic b.p was reduced by more than 10% then the sentence below by 8% ,please clarify.

In discussion : instead of the results of 2018 study write  the name of the author and hos findings.

Need literature that support or contradict your findings that pt improved mobility.the same for heart rate and O2 saturation. 

Other limitation in the study is that your sample lifestyle was not considered including the effect of their work on their back .

As you mentioned in the conclusion that your study open chance for stidying cardiovascular parameters you may need to add more literature in discussion to discuss effect of PT both electro and exercise on cardo vascular parameters. 

Reviewer 3 Report

Comments and Suggestions for Authors

Number under 10 must be written in letters. Line 16.

Once Low back pain abbreviation is defined, always use it. Line 46...

G1 and G2 are not defined.

Oswestry scale is not the name of anu scale. Check and correct it.

 The VAS between T1-T2, T2-T3, and T1-T3, does not have the correct sign

ODI is not defined before, and it is not the abbreviation of disability